# Diallyl Trisulfide (DATS) Suppresses AGE-Induced Cardiomyocyte Apoptosis by Targeting ROS-Mediated PKCδ Activation

**DOI:** 10.3390/ijms21072608

**Published:** 2020-04-09

**Authors:** Dennis Jine-Yuan Hsieh, Shang-Chuan Ng, Ren-You Zeng, Viswanadha Vijaya Padma, Chih-Yang Huang, Wei-Wen Kuo

**Affiliations:** 1School of Medical Laboratory and Biotechnology, Chung Shan Medical University, Taichung 402, Taiwan; djh@csmu.edu.tw; 2Clinical Laboratory, Chung Shan Medical University Hospital, Taichung 402, Taiwan; 3Department of Biological Science and Technology, College of Biopharmaceutical and Food Sciences, China Medical University, Taichung 404, Taiwan; shangchuan_622@hotmail.com (S.-C.N.); renyou@gmail.com (R.-Y.Z.); 4Translational Research Laboratory, Department of Biotechnology, School of Biotechnology and Genetic Engineering, Bharathiar University, Coimbatore 641046, Tamil Nadu, India; padma.vijaya@gmail.com; 5Graduate Institute of Biomedical Sciences, China Medical University, Taichung 404, Taiwan; cyhuang@mail.cmu.edu.tw; 6Cardiovascular and Mitochondrial Related Disease Research Center, Hualien Tzu Chi Hospital, Buddhist Tzu Chi Medical Foundation, Hualien 970, Taiwan; 7Center of General Education, Buddhist Tzu Chi Medical Foundation, Tzu Chi University of Science and Technology, Hualien 970, Taiwan; 8Department of Medical Research, China Medical University Hospital, China Medical University, Taichung 404, Taiwan; 9Department of Biotechnology, Asia University, Taichung 413, Taiwan

**Keywords:** DATS, AGE, PKCδ, cardiomyocyte, apoptosis

## Abstract

Chronic high-glucose exposure results in the production of advanced glycation end-products (AGEs) leading to reactive oxygen species (ROS) generation, which contributes to the development of diabetic cardiomyopathy. PKCδ activation leading to ROS production and mitochondrial dysfunction involved in AGE-induced cardiomyocyte apoptosis was reported in our previous study. Diallyl trisulfide (DATS) is a natural cytoprotective compound under various stress conditions. In this study, the cardioprotective effect of DATS against rat streptozotocin (STZ)-induced diabetic mellitus (DM) and AGE-induced H9c2 cardiomyoblast cell/neonatal rat ventricular myocyte (NRVM) damage was assessed. We observed that DATS treatment led to a dose-dependent increase in cell viability and decreased levels of ROS, inhibition of PKCδ activation, and recuded apoptosis-related proteins. Most importantly, DATS reduced PKCδ mitochondrial translocation induced by AGE. However, apoptosis was not inhibited by DATS in cells transfected with PKCδ-wild type (WT). Inhibition of PKCδ by PKCδ-kinase-deficient (KD) or rottlerin not only inhibited cardiac PKCδ activation but also attenuated cardiac cell apoptosis. Interestingly, overexpression of PKCδ-WT plasmids reversed the inhibitory effects of DATS on PKCδ activation and apoptosis in cardiac cells exposed to AGE, indicating that DATS may inhibit AGE-induced apoptosis by downregulating PKCδ activation. Similar results were observed in AGE-induced NRVM cells and STZ-treated DM rats following DATS administration. Taken together, our results suggested that DATS reduced AGE-induced cardiomyocyte apoptosis by eliminating ROS and downstream PKCδ signaling, suggesting that DATS has potential in diabetic cardiomyopathy (DCM) treatment.

## 1. Introduction

Diabetes mellitus (DM) is the most lethal noncommunicable disease worldwide, with the global prevalence expected to rise to 366 million by 2030 [1,2]. Elevated levels of blood glucose is the major feature of both type I and type II DM [3]. Oxidative stress induced by high glucose is defined as the imbalance of ROS production and scavenging, leading to cellular damage and apoptosis [4]. Growing evidence suggests that high levels of glucose and excessive generation of reactive oxygen species (ROS) contribute to the development and progression of various diabetic complications, such as ischemic heart disease, myocardial infarction, and cardiomyopathy [5]. Thus, finding an ideal candidate to suppress ROS generation and apoptosis is important.

Hyperglycemia in DM accelerates the formation of advanced glycation end-products (AGEs) through various mechanisms, with one being the nonenzymatic glycation process with intra- and extracellular proteins [6]. Glucose reacts with protein amino (NH_2_) groups and forms reversible Amadori products. Then, rearrangement reactions occur to form AGEs, which remain irreversibly bound to proteins [7]. AGE-modified molecules can directly cross-link proteins and impair their function and structure [8]. The interactions between AGEs and cell organelles are highly correlated with the pathogenesis of several complications associated with diabetes and cellular aging; thus, the role of AGEs is an increasingly important issue in various disorders, including cardiovascular diseases [9]. Furthermore, the actions of AGEs are mediated by the receptor for AGE (RAGE), a multiligand receptor that exerts its biological effects by binding to ligands, including AGEs [10].

PKCs are a family of serine/threonine protein kinases that are activated through different stimuli and play an important role in regulating various functions in cells, such as proliferation, apoptosis, growth, and differentiation [11,12]. Based on the domain structure and regulation mechanism, twelve members of the PKC family can be divided into three groups, namely, conventional PKCs (cPKCs), novel PKCs (nPKCs), and atypical PKCs (aPKCs). PKCδ belongs to the nPKC subfamily, which is calcium-insensitive but diacylglycerol-responsive [13]; it is ubiquitously expressed in many tissues, including hearts [14]. PKCδ is activated and upregulated in various types of cardiovascular diseases, such as myocardial infarction [15], hypertrophy [16], and ischemia–reperfusion [16]. PKCδ can be activated by H_2_O_2_ and high levels of glucose through increased mitochondrial ROS and enhanced oxidative stress [17,18,19]. Moreover, mitochondrial translocation of PKCδ contributes to ischemia–reperfusion injury by inhibiting ATP production and promoting mitochondrial ROS generation, cytochrome *c* release, caspase activation, and apoptosis induction [20]. Our previous study linked the activation of PKCδ and mitochondrial dysfunction in cardiomyocytes to ROS upregulation in response to AGE exposure. We demonstrated that among the different PKC isoforms, PKCδ was a critical regulator of mitochondrial dynamics and increased mitochondrial fragmentation and reduced their biological functions, leading to cardiac failure [21]. Therefore, the inhibition of PKCδ signaling pathways may be a good strategy to reduce AGE-induced ROS-mediated mitochondrial dysfunction and apoptosis in cardiomyocytes.

Garlic is a common food ingredient in our daily life and, due to its health benefits, it has become a dietary supplement [22]. Garlic is also well known in medicine. Many studies suggest that garlic exerts physiological effects, such as antihypertension, anticancer, antithrombotic, and antioxidant effects [23,24]. Allyl sulfur compounds abundant in garlic can convert into allicin by allinase when the garlic tissue is crushed [25]. Allicin is an unstable compound and further converts into organosulfur compounds, such as diallyl sulfide (DAS) [26], diallyl disulfide (DADS), diallyl trisulfide (DATS), and methyl allyl trisulfide. These compounds generally exist in proportions of 3.77% DAS, 40.83% DADS, 38.93% DATS (also known as allitridin), and 7.17% methyl allyl trisulfide [27]. A previous study related to garlic ordered the antioxidant potential of these compounds as DATS > DADS > DAS [28]. Some studies suggested that DATS, which contains three sulfur atoms, can increase ROS generation to inhibit prostate and breast cancer cell growth and cause cell cycle arrest in the G2 phase. However, DATS is a source of H_2_S, which is a beneficial cardioprotective agent with antiapoptotic effects on cardiac cells [28]. Regarding cardioprotection, we determined whether AGE-induced PKCδ activation and ROS-mediated apoptosis could be attenuated by DATS.

## 2. Results

### 2.1. AGE Induced Cardiac PKCδ Protein Expression, Phosphorylation, and Apoptosis in a Dose- and Time-Dependent Manner

Firstly, we examined cardiac survival- and apoptosis-related proteins in AGE-treated H9c2 cardiomyoblast cells. In addition to the upregulation of p-PKCδ, we observed that survival-related proteins, including p-Akt, were reduced and apoptosis-related proteins, including cleaved caspase-9, cleaved caspase-3, and cytochrome *c*, were increased in cardiac cells following AGE exposure in a dose- and time-dependent manner (Figure 1A,B).

### 2.2. Inhibitory Effect of DATS on AGE-Induced Cardiac Apoptosis and PKCδ Mitochondrial Translocation

Next, we performed an MTT assay to assess the cytotoxic effects of DATS on H9c2 cardiomyoblast cells. The results showed that DATS did not induce cell damage when the concentration was lower than 10 µM. The chosen dose of DATS for the following experiments was not greater than 10 µM (Figure 2A). By exploring the effects of DATS on survival markers, we found that DATS treatment increased p-Akt protein levels, indicating that DATS may maintain cardiomyoblast growth and apoptosis inhibition (Figure 2B). Our prior studies demonstrated that PKCδ activation was involved in ROS-mediated mitochondrial dysfunction and apoptosis in response to AGE exposure [21]. To explore whether DATS could inhibit PKCδ-dependent apoptosis in AGE-exposed H9c2 cells, we performed Western blotting to examine the related proteins. The Western blot analysis results showed an increase in the survival pathway-related protein p-Akt and a decrease in PKCδ activation and apoptosis-related proteins, such as cleaved caspase-9, cleaved caspase-3, and cytochrome *c*, after DATS treatment (Figure 2D). Furthermore, the data of the MTT assay following the treatment with different concentrations of DATS at 1, 5, and 10 μM for 24 h in cardiomyoblast cells exposed to AGE at 250 μg/mL demonstrated that DATS treatment significantly reduced the apoptotic effect of AGE (Figure 2C). Mitochondrial PKCδ activation was examined using a mitochondrial and cytosol separation kit and Western blotting. H9c2 cells were incubated with DATS in the presence of AGE. As shown in Figure 2E, mitochondrial translocation of PKCδ increased in the AGE-treated H9c2 cells and was inhibited by DATS treatment in a dose-dependent manner.

### 2.3. Antioxidant Effect of DATS on AGE-Induced Cardiac ROS Generation

To examine ROS production, MitoSOX^TM^ Red Mitochondrial Superoxide Indicator (Figure 3B) and 2′,7′-dichlorofluorescin diacetate (DCF-DA) (Figure 3A) were used to detect the total and mitochondrial ROS. The results of the flow cytometry analysis showed that DATS reduced ROS generation induced by AGE in a dose-dependent manner. Furthermore, since reduced mitochondrial membrane potential (MMP) can result in a cascade of apoptotic processes, we evaluated whether DATS could decrease MMP when enhanchaed by AGE exposure to cause apoptosis; JC-1 staining was performed to stain the MMP. Normal mitochondria were exhibited in red fluorescence, while AGE exposure increased green fluorescence, indicating a loss of MMP, which was closely related to the development of apoptosis. In contrast, treatment with DATS largely enhanced red fluorescence, suggesting increased MMP (Figure 3C). The results indicated that DATS may protect against AGE-induced oxidative stress injury by eliminating ROS production and increasing MMP in H9c2 cells.

### 2.4. Cardiac PKCδ-Dependent Apoptosis Induced by AGE is Inhibited by GFP-PKCδ-KD

As shown in (Figure 4A), following GFP-PKCδ-wild type (WT) overexpression in cardiac cells, the related proteins were measured. GFP-PKCδ-WT plasmid-transfected cells exhibited increased PKCδ expression and apoptosis-related markers, indicating successful PKCδ-WT plasmid transfection. To examine the potential involvement of PKCδ in AGE-induced apoptosis, the effect of the selective kinase-deficient PKCδ (GFP-PKCδ-KD) mutant was overexpressed and apoptosis was evaluated. First, as shown in (Figure 4B), AGE significantly increased the proteins levels of p-PKCδ, cleaved caspase-9, cleaved caspase-3, and cytochrome *c*, which were dose-dependently inhibited by GFP-PKCδ-KD overexpression, indicating that overexpressed PKCδ-KD functions as a PKCδ-dominant-negative mutant. For the following experiments, 2 µg of GFP-PKCδ-WT or GFP-PKCδ-KD was selected as the dose.

### 2.5. DATS Suppresses Cardiac Apoptosis by Inhibiting PKCδ Activation and its Downstream Apoptosis-Related Proteins Following AGE Exposure

To confirm previous experimental results (Figure 2D) demonstrating that DATS inhibits AGE-induced PKCδ activation, the selective PKCδ inhibitor, rottlerin, and PKCδ-KD overexpression were used to identify whether the antiapoptotic effect of DATS was mediated via PKCδ activation inhibition. As shown in Figure 5A,B, the levels of p-PKCδ, cleaved caspase-9, cleaved caspase-3, and cytochrome *c* were upregulated, while the survival protein p-Akt was decreased in both AGE-exposed and GFP-PKCδ-WT transfection models. Surprisingly, we found that in addition to DATS, both rottlerin and GFP-PKCδ-KD inhibited PKCδ activation and attenuated PKCδ-dependent apoptosis. To further confirm that DATS could inhibit AGE-induced PKCδ-mediated apoptosis, we exposed NRVM cells (Figure 5C) and H9c2 cells (Figure 5D) to AGE and treated them with DATS. Concurrently, we overexpressed the PKCδ-WT plasmid and found that the antiapoptotic effect of DATS treatment on AGE exposure could be reversed. AGE-induced apoptosis was not observed in GFP-PKCδ-KD-transfected cells, indicating the involvement of PKCδ in regulating cardiac cell apoptosis following AGE exposure.

### 2.6. DATS Suppresses Cardiac Apoptosis by Inhibiting Activation of PKCδ and Apoptosis-Related Signaling Pathways in STZ-Induced Diabetic Rats

Diabetes was induced in rats by intraperitoneal (IP) injection of streptozotocin (STZ) at a dose of 65 mg/kg and oral gavage with DATS at dose 40mg/kg BW every other day for 16 days (Figure 6A). We found that treatment with STZ upregulated the protein levels of p-PKCδ, cleaved caspase-3, and cleaved caspase-9, and downregulated survival markers, such as p-Akt, in diabetic rat hearts. However, we examined p-PKCδ levels via immunohistochemistry (IHC) and found that cardiac p-PKCδ levels were significantly increased in DM rats, with DATS treatment reducing p-PKCδ levels (Figure 6B). Furthermore, the changes in the levels of these apoptosis-related proteins and p-PKCδ were significantly reversed in STZ-treated rats following DATS administration (Figure 6C). By echocardiographic analysis, we previously demonstrated that cardiac functions in STZ-treated rats were impaired, with this impairment being effectively ameliorated following DATS treatment, indicating that DATS attenuated PKCδ-mediated apoptotic signaling pathways, thus improving cardiac function in diabetic rats [29].

## 3. Discussion

In this study, we investigated the anti-apoptotic role of DATS through the inhibition of PKCδ activation in response to AGE in both H9c2/NRVM cells and animal models. The results of this study can be summarized in four main points: (1) AGE-induced ROS-mediated PKCδ activation and apoptosis were attenuated following DATS treatment; (2) DATS treatment reduced intracellular and mitochondrial ROS induced by AGE; (3) PKCδ-mediated apoptosis pathways induced by AGE were highly suppressed after DATS treatment, and the effects were reversed following PKCδ-WT overexpression in H9c2 cardiomyoblasts and NRVMs; and (4) the levels of p-PKCδ and proapoptotic proteins, such as cleaved caspase-3, were highly increased in STZ-induced DM rats and suppressed in response to DATS administration. The possible signaling pathways for DATS-mediated inhibition of PKCδ activation are shown in Figure 7, indicating that DATS protected cardiac cells against apoptosis by reversing ROS-mediated PKCδ activation, thereby leading to cardiomyocyte apoptosis. Our findings suggested that DATS administration could serve as an alternative treatment against AGE-induced apoptosis. 

Accumulated evidence suggests that DATS treatment attenuates stress-induced mitochondria-mediated apoptosis. In endothelial cells, dynamin-related protein 1 (Drp1) triggered mitochondria dysfunction, which was reversed under DATS administration [30]. Silent information regulator l (SIRT1), which acts as the regulator of mitochondrial biogenesis, was significantly enhanced by DATS in myocardial ischemia–reperfusion (MI/R) injury models [31]. Therefore, reviving mitochondrial damage may be another way that DATS can protect cardiac cells from apoptosis. In our study, we found that a DATS concentration of less than 10 µM did not exhibit any cytotoxicity toward H9c2 cardiomyoblast cells (Figure 2A). Furthermore, p-Akt levels were significantly higher upon DATS treatment, indicating its survival-promoting role (Figure 2B). The enhancement of cell viability by DATS was identified by MTT assay after AGE exposure (Figure 2C). Apoptosis can be initiated by both intrinsic and extrinsic pathways [32]. AGE-induced cardiac apoptosis belongs to the intrinsic apoptosis pathway, as demonstrated in our previous study. AGE upregulated p-PKCδ and apoptosis-related markers, including cleaved caspase-9, cleaved caspase-3, and cytochrome *c*, and downregulated p-AKT. However, PKCδ activation and apoptosis-related markers were repressed, while p-Akt levels were reversed following DATS treatment (Figure 2D). Additionally, the activation of PKCδ results in its translocation to different cellular organelles, such as mitochondria, Golgi, nucleus, and endoplasmic reticulum, where it exerts different biological functions [26,33]. PKCδ mitochondrial translocation can be initiated by proapoptotic pathways [20]. Furthermore, phosphorylation of Thr505 in the catalytic domain of PKCδ was also reported to occur in response to oxidative stress [13,34]. Our results showed that PKCδ-activated mitochondrial translocation was diminished by DATS treatment in a dose-dependent manner and retained in the cytosol (Figure 2E). Taken together, our results suggested that PKCδ may be an important target for DATS regarding H9c2 cardiomyoblast apoptosis inhibition.

Overproduction of ROS is highly correlated with the development of diabetic complications [35]. Mitochondria are intracellular organelles that produce adenosine triphosphate (ATP) through the TCA cycle, coupling oxidative phosphorylation and electron transfer. During energy generation, ROS are produced as side products of respiration in human cells [36]. The majority of basal cellular ROS (approximately 80%) are produced in mitochondria undergoing high respiratory activity, such as cardiomyocytes [37]. Our results showed that AGE caused increased production of intracellular and mitochondrial ROS and reduced MMP in H9c2 cardiomyoblast cells, but these effects were reversed following DATS (1–10 µM) treatment (Figure 3A,B). Therefore, eliminating ROS may be a potential way of preventing cardiac cell apoptosis.

To investigate the involvement of PKCδ in response to AGE, PKCδ-WT (Figure 4A) and PKCδ-KD (Figure 4B) constructs were overexpressed to assess their apoptotic effects in H9c2 cardiomyoblast cells. Furthermore, overexpression of PKCδ-KD, which is a PKCδ^K376R^ domain mutant (Figure 5A), or treatment with rottlerin, a selective PKCδ inhibitor (Figure 5B), resulted in apoptosis inhibition. Notably, DATS treatment had the same effect on the inactivation of PKCδ as PKCδ-KD and rottlerin. Interestingly, overexpression of PKCδ reversed DATS-mediated inhibition of p-PKCδ, cleaved caspase-9, cleaved caspase-3, and cytochrome *c* and increased levels of the survival marker p-AKT in both H9c2 and NRVM cells exposed to AGE, indicating that DATS exerts its antiapoptotic effects by regulating PKCδ signaling (Figure 5C,D). Taken together, our study results demonstrated that DATS protects against AGE-induced cardiomyocyte death by inactivating PKCδ signaling pathways.

AGE-induced oxidative stress was reported in the development of diabetic complications [38]. The inhibition of ROS formation and PKCδ activation may be a potential way to reduce cardiac apoptosis. In conclusion, we demonstrated that DATS exerts its cardioprotective effects against cardiac cell apoptosis by suppressing oxidative stress-induced PKCδ activation in both in vivo and in vitro studies, strongly suggesting that DATS could be a potential therapeutic agent for diabetic complications, such as diabetic cardiomyopathy.

## 4. Materials and Methods 

### 4.1. Cell Culture

The rat cardiomyoblast H9c2 cells (rat embryonic cardiac myoblasts; ATCC CRL-1446, VA, USA) were grown in low glucose Dulbecco’s modified essential medium (Sigma-Aldrich D6046, MO, USA) with 10% HyClone™ Cosmic Calf™ Serum (U.S.) and 1% penicillin (Corning^®^, 30-002-CI), and incubated in a humidified incubator with 5% CO_2_ at 37 °C. 

### 4.2. Neonatal Rat Ventricular Myocyte (NRVM) Primary Culture

Neonatal rat ventricular cardiomyocytes (NRVMs) were isolated and cultured using a Neonatal Rat/Mouse Cardiomyocyte Isolation Kit (Cellutron Life Technology, Baltimore, MD, USA, NC-6031). Briefly, hearts from one- or two-day-old Sprague Dawley rats (BioLASCO, Taipei, Taiwan) were separated and transferred to a sterile beaker. Each heart was digested and stirred in the beaker at 37 °C for 15 min. The supernatant was then transferred to a new sterile tube and spun at 1200 rpm for 1 min. The cell pellets were then resuspended in D3 buffer and pre-plated for 1 h by seeding on an uncoated plate at 37 °C in a CO_2_ incubator to select cardiac fibroblasts. The unattached cells were transferred to plates that were pre-coated with NS medium supplemented with 10% FBS (Cellutron Life Technology, Baltimore, MD, USA, m-8031). Ventricular cardiomyocytes were incubated in NS medium. The cardiomyocyte cultures were ready for experiments 48 h after the initial plating.

### 4.3. Glucose-Derived AGE Preparation

AGE was prepared according to methods described previously [11,39,40]. Briefly, AGE was prepared by incubating bovine serum albumin (A8806, BSA, fraction V, fatty acid-free, endotoxin-free; Sigma Chemical, 100 mg/mL) with D-glucose (1 M) in Dulbecco’s phosphate-buffered saline (DPBS, 21600–010; Gibco) for 12 weeks at 37 °C. Unmodified BSA was prepared under the same conditions without glucose as a control. The fluorescence of the supernatant was determined (excitation 370 nm, emission 440 nm) using LJL Biosystems (Analyst^TM^ AD, Sunnyvale, CA, USA), which confirmed the higher intensity of AGEs in AGE-modified BSA than in unmodified BSA in our previous paper [21]. The AGE solution was filter sterilized by a 0.8 μM Millex GP filter unit (Millipore, Billerica, MA, USA).

### 4.4. Transient Transfection of Plasmid DNA

The pEGFP N1-PKCδ (GFP-PKCδ-WT) and pEGFP N1-PKCδ^K376R^ (GFP-PKCδ-KD; PKCδ kinase dominant-negative mutant) plasmids were provided by the Department of Life Science and the Graduate Institute of Biomedical Sciences, National Chung Hsing University, Dr. H.C. Chen’s laboratory [41]. The cells were grown to approximately 80% confluence on the day of transfection. Plasmids were transfected into H9c2 or NRVM cells for 24 h by the PureFection Transfection Reagent (System Biosciences, Inc., Mountain View, California, USA), according to the manufacturer’s protocol.

### 4.5. Western Blot Analysis

Cells were lysed in RIPA buffer prepared as indicated (50 mM Tris (pH 7.5), 150 mM NaCl, 1% NP-40, 0.1% SDS, and 0.5% sodium deoxycholate) for 1 h and centrifuged at 12,000 rpm for 15 min. The Bradford method was used to quantify the proteins, and 10 μg of proteins for each sample were separated by (6–12%) SDS-PAGE and transferred to PVDF membranes (GE Healthcare, Amersham, UK ) [42]. The membranes were blocked with 5% nonfat milk for 1 h and probed with 1:1000 diluted specific primary antibodies at 4 °C overnight; secondary antibodies included anti-162 rabbit IgG (Sigma-A0545-1ML), anti-mouse IgG (Sigma-A9044-2ML), or anti-goat IgG (Sigma-163 A5420-1ML), which were subsequently used at room temperature for 1 h on an orbital shaker. The membranse were then analyzed using chemiluminescence (ECL) reagent by an AlphaImager2200 digital imaging system (Digital Imaging System, Commerce, CA, USA). The results were analyzed and quantified using Image J software (NIH, MD, USA).The following antibodies were used in this study: anti-Akt1 (sc-5298-Santa Cruz), anti-p-Akt1/2/3Ser473 (sc-7985; Santa Cruz), anti-β-actin (C4) (sc-47778; Santa Cruz), anti-cleaved caspase-3 (#9664; Cell Signaling), anti-cleaved caspase-9 (#9507; Cell Signaling), anti-COX IV (#11967; Cell Signaling), anti-cytochrome *c* (sc-13560; Santa Cruz), anti-GAPDH (6C5) (sc-32233; Santa Cruz), anti-PKCδ (#2058; Cell Signaling), and anti-p-PKCδThr 505/507 (bs-3727R; Bioss).

### 4.6. MTT Assay

A 3-(4,5-dimethylthiazol-2-yl)-2,5-diphenyltetrazolium bromide (Sigma-M2128) MTT assay was used to determine cell viability. H9c2 cells was seeded in 24-well plates at a density of 2 × 10^4^ cells/well and allowed to attach overnight at 37 °C in a CO_2_ incubator. AGEs were prepared as described above, and DATS was dissolved in DMSO. H9c2 cells were treated for 1 h with AGEs (250 μg/mL) and then co-treated with DATS (ab141926) at different concentrations (1, 5, or 10 μM) for 23 h or treated with DATS (1, 3, 5, 7, 10, 15, or 20 µM) alone. After the treatment period, 500 µL of MTT (5 mg/mL) was added to each well and incubated for 3 h at 37 °C in a CO_2_ incubator. The formazan crystals that formed were dissolved by adding 200 μL of DMSO. The optical density (OD) values of samples were measured at 580 nm using Epoch™ Microplate Spectrophotometer (BioTek, Winooski, VT, USA). Results were expressed as percentage of cell viability. 

### 4.7. Intracellular and Mitochondrial Reactive Oxygen Species (ROS) Production

Intracellular and mitochondrial ROS were examined by flow cytometry using DCFH-DA (Sigma-D6883) and Molecular Probes MitoSOX™ (Invitrogen™-M36008) Red mitochondrial superoxide indicator. Briefly, cells were incubated with (5 μM) DCFH-DA at 37 °C for 30 min. Furthermore, mitochondrial ROS were detected by red mitochondrial superoxide indicator. Cells were incubated with MitoSOX^TM^ (2 μM) at 37 °C for 20 min. H_2_O_2_ (200 µM) was used as a positive control. All samples were analyzed with a BD FACS Canto M II flow cytometer (Becton–Dickinson, Franklin Lakes, NJ) to identify intracellular and mitochondrial ROS production. The analysis was repeated 3 times for each treatment.

### 4.8. Measurement of Mitochondrial Membrane Potential (MMP) 

Mitochondrial membrane potential (MMP) was examined using a Mitochondria Staining Kit to detect mitochondrial potential changes (Sigma-CS0390) according to the manufacturer’s instructions. Briefly, H9c2 cells were seeded on glass cover slips in 4-well plates for 24 h. After treatment, cells were incubated with JC-1 at 37 °C for 20 min and then washed twice with JC-1 dyeing buffer. The cells dyed with JC-1 were assessed using a fluorescence microscope (Olympus, Tokyo, Japan). The red fluorescence of the aggregated JC-1 represented the intact mitochondria, while green fluorescence of monomeric JC-1 represented the disrupted mitochondria.

### 4.9. Isolation of Mitochondria

Mitochondria were isolated using a mitochondria isolation kit (Thermo Scientific, Rockford, IL, USA). Briefly, H9c2 cells were washed with pre-chilled PBS and harvested by trypsinization. The cells were suspended in 800 μL of mitochondria isolation reagent A and incubated for 2 min on ice. Samples totaling 10 μL of reagent B were then added to the cells, followed by incubation for 5 min. After incubation, 800 μL of reagent C was added to the tube, followed by centrifugation at 700g for 10 min at 4 °C. Both the pellets (mitochondria fraction) and supernatants (cytosol fraction) were retained for further analysis. Mitochondrial and cytosol protein concentrations were determined using the Bradford method as the standard and analyzed by Western blotting. 

### 4.10. Animals and Treatment Protocol

All animal experiments were performed in accordance with the Guide for the Care and Use of Laboratory Animals (National Institutes of Health Publication No. 85-23, revised 1996) under the protocol approved by the Animal Research Committee of China Medical University, Taichung, Taiwan (CMUIACUC-2018-129-1). Four-week-old male Wistar rats were obtained from the National Animal Breeding and Research Center (Taipei, Taiwan). The rats were housed at a constant temperature (22 °C) on a 12-h light/dark cycle with food and tap water. After one week of adaptation to the environment, diabetes was induced by injecting streptozotocin (STZ, 65 mg/kg body weight in citrate buffer, pH 4.5) into a lateral tail vein. Control animals were injected with the same volume of vehicle. After three days of injection, blood glucose was measured with the Accu-Check Compact kit (Roche Diagnostics GmbH, Mannheim, Germany). Animals with successful hyperglycemia induction, measured by blood glucose levels of <200 mg/dL, were randomly assigned to two groups and received 40 mg/kg body weight DATS or a corn oil vehicle of 2 mL/kg body weight by gavage every other day for 16 days. The rats were divided into three groups, namely, the control groups (C, *n* = 6), the STZ-induced diabetic mellitus group (DM, *n* = 6), and the STZ-induced diabetic mellitus + DATS treatment group (DM + DATS, *n* = 6). Sixteen days after treatment, all animals were anesthetized and echocardiography was performed to check cardiac function. Impairment of cardiac function analyzed by echocardiography indicated successful induction of diabetic cardiomyopathy. All animals were then sacrificed and the hearts were removed for further analysis. The DATS doses used in this study were according to a previous study where DM rats received 40 mg/kg body weight DATS and showed improvement of cardiac function [27].

### 4.11. Immunohistochemistry

The slides were immunostained with anti-p-PKCδ T505/507 antibodies (1:100 dilution; bs-208 3727R; Bioss) using the Ultra Vision LP Detection System (Vector Laboratories, California, USA) according to the manufacturer’s instructions. The tissue sections were deparaffinized and rehydrated. After three washes in PBS, hydrogen peroxide blocking buffer was used to block endogenous peroxidase activity for 10 min. Ultra V Block was incubated for 5 min at room temperature to block nonspecific background staining. Then, the sections were incubated with a primary antibody against p-PKCδ T505/507 (1:100 dilutions; bs-3727R; Bioss) at 4 °C overnight in a moist chamber. After incubation with horseradish peroxidase for 20 min at 37 °C, these antibodies were located using a universal secondary antibody formulation conjugated to an enzyme-labeled HRP polymer. After staining with an appropriate substrate/chromogen for 5 min, the slide was counterstained with Harris hematoxylin, and cover slipped with permanent mounting media (Sigma Chemical, Missouri, USA). The expression of p-PKCδ was then detected using microscopy (Olympus, Tokyo Japan). 

### 4.12. Statistical Analysis

Each experiment was repeated at least three times. Statistical analyses were performed using GraphPad Prism version 5 (GraphPad Software, Inc.). The data were analyzed by one-way ANOVA. Significance between the individual means was determined by Tukey’s test. Imaging results were quantified by ImageJ and processed with Adobe Photoshop (Adobe Systems, Inc, California, USA). *p* < 0.05 was considered to indicate a statistically significant difference.

## Figures and Tables

**Figure 1 ijms-21-02608-f001:**
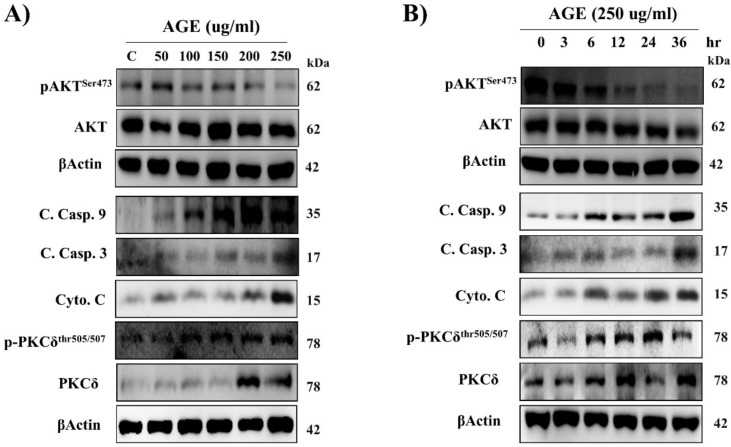
Advanced glycation end-product (AGE) induced cardiac PKCδ protein expression, phosphorylation, and apoptosis in a dose- and time-dependent manner. H9c2 cells were treated (**A**) with different doses of AGE (50, 100, 150, 200, and 250 μg/mL) for 24 h and (**B**) for different time periods (0, 3, 6, 12, 24, and 36 h) with AGE at 250 μg/mL. Protein levels were analyzed by Western blotting and β-actin was used as a loading control.

**Figure 2 ijms-21-02608-f002:**
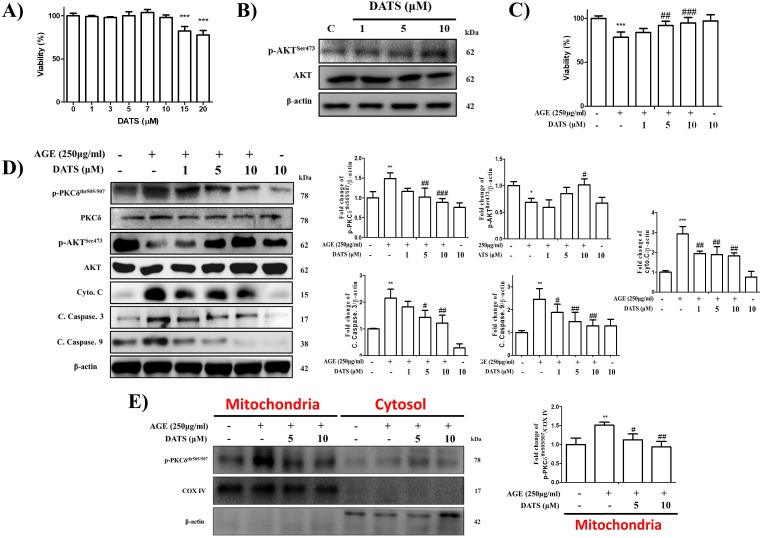
Inhibitory effect of diallyl trisulfide (DATS) on AGE-induced cardiac apoptosis and PKCδ mitochondrial translocation. (**A**) Cells were treated with DATS at different concentrations (0, 1, 3, 5, 7, 10, 15, and 20 µM) for 24 h. Cell viability was examined using an MTT assay. (**B**) Western blotting was performed to check the effects of DATS on the expression of the survival marker p-AKT. (**C**) Inhibition of AGE-induced apoptosis was examined using an MTT assay to assess cell viability following DATS treatment. (**D**,**E**) Following treatment with DATS (5 and 10 µM) after 1 h, AGE (250 µg/mL) was added to induce apoptosis for 24 h. The levels of p-PKCδ protein and apoptosis-related markers were assessed by Western blotting. p-PKCδ expression levels in the cytosol and mitochondria of H9c2 cardiomyoblast cells were seperated using a mitochondria and cytosol separation kit and analyzed by Western blotting. β-actin and COX IV were used as a loading control. Values shown are means ± SD. Quantification of the results is shown (*n* = 3); * *p* < 0.05, ** *p* < 0.01, and *** *p* < 0.001 versus control cells; ^#^
*p* < 0.05, ^##^
*p* < 0.01, and ^###^
*p* < 0.001 versus AGE-treated cells.

**Figure 3 ijms-21-02608-f003:**
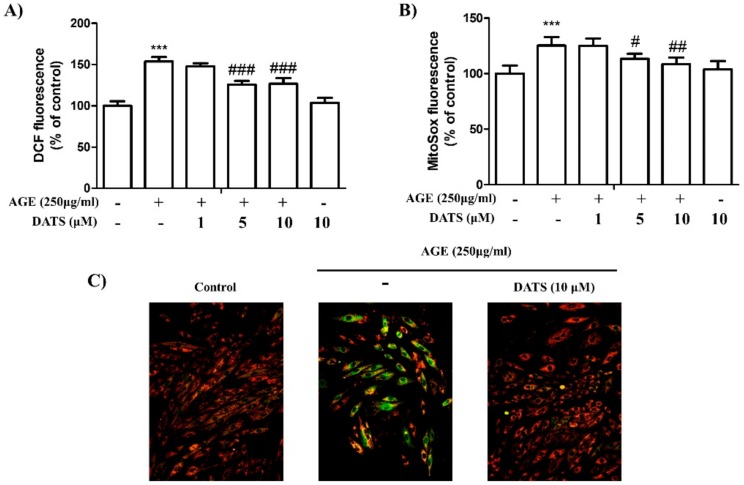
Antioxidant effect of DATS on AGE-induced cardiac reactive oxygen species (ROS) generation. Following treatment with DATS (1, 5, and 10 μM) after 1 h, AGE (250 μg/mL) was added to induce ROS generation for 24 h. (**A**) 2’,7’-dichlorofluorescin diacetate (DCF-DA) and (**B**) MitoSOX^TM^ Red Mitochondrial Superoxide Indicator were used to detect intracellular and mitochondrial ROS production. (**C**) Mitochondrial membrane potential (MMP) was determined via JC-1 staining. Values shown are means ± SD. Quantification of the results is shown (*n* = 3); *** *p* < 0.001 versus control cells; ^#^
*p* < 0.05, ^##^
*p* < 0.01, and ^###^
*p* < 0.001 versus AGE-treated cells.

**Figure 4 ijms-21-02608-f004:**
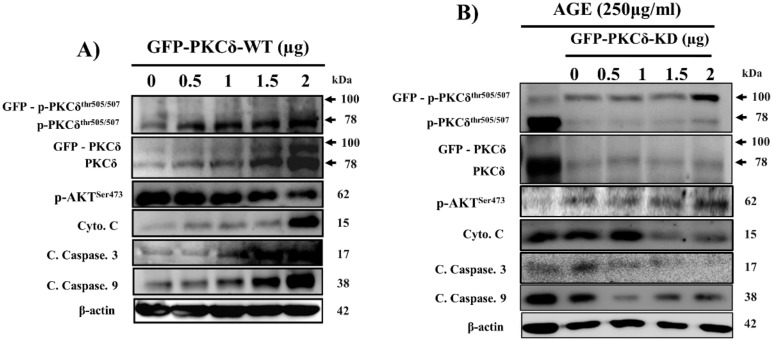
Cardiac PKCδ-dependent apoptosis induced by AGE is inhibited by selective kinase-deficient PKCδ (GFP-PKCδ-KD). (**A**) Cells were transfected with GFP-PKCδ-wild type (WT) at the indicated doses. (**B**) H9c2 cells were exposed to AGE (250 µg/mL) and transfected with GFP-PKCδ-KD at the indicated doses. Protein levels were analyzed by Western blotting and β-actin was used as a loading control.

**Figure 5 ijms-21-02608-f005:**
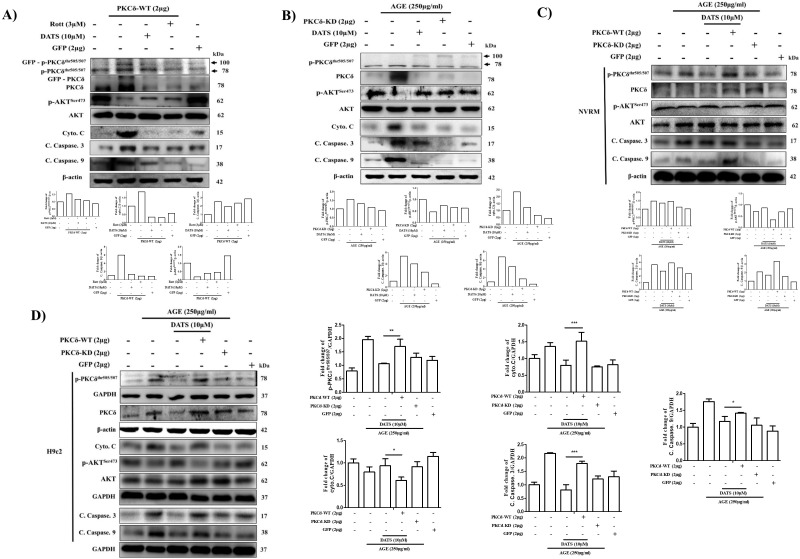
DATS suppresses cardiac apoptosis by inhibiting PKCδ activation and its downstream apoptosis-related proteins following AGE exposure. (**A**) Cells were exposed to AGE (250 μg/mL) followed by DATS (10 µΜ) treatment or PKCδ-KD (2 μg) transfection. (**B**) Cells were transfected with PKCδ-WT plasmid followed by DATS or rott treatment. (**C**) NRVM or (**D**) H9c2 cells were exposed to AGE (250 μg/mL) with or without PKCδ-KD transfection or treatment with DATS in the presence or absence of PKCδ-WT transfection. Protein levels were analyzed by Western blotting and β-actin and GAPDH was used as loading controls. Values shown are means ± SD. Quantification of the results is shown (*n* = 3); * *p* < 0.05, ** *p* < 0.01, and *** *p* < 0.001 versus AGE + DATS-treated cells.

**Figure 6 ijms-21-02608-f006:**
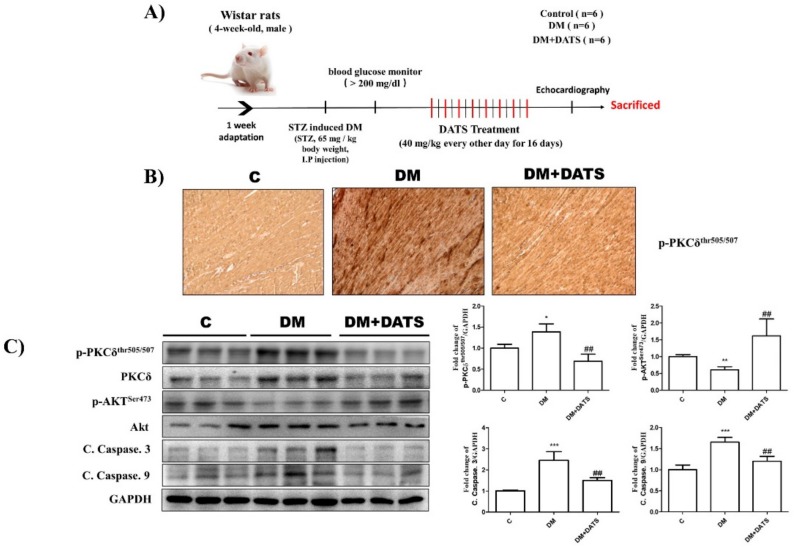
DATS suppresses cardiac apoptosis by inhibiting the activation of PKCδ and apoptosis-related signaling pathways in streptozotocin (STZ)-induced diabetic rats. (**A**) The schematic procedure of STZ-induced diabetes mellitus (DM) and DATS treatment. (**B**) Cardiac expression of phosphorylated PKCδ was examined by immunohistochemistry analysis. (**C**) Western blot analysis of the phosphorylation levels of PKCδ and apoptosis-related proteins in STZ-induced diabetic rat hearts. Values shown are means ± SD. Quantification of the results is shown (*n* = 3); * *p* < 0.05, ** *p* < 0.01 and *** *p* < 0.001 versus control and ^##^
*p* < 0.01 versus DM.

**Figure 7 ijms-21-02608-f007:**
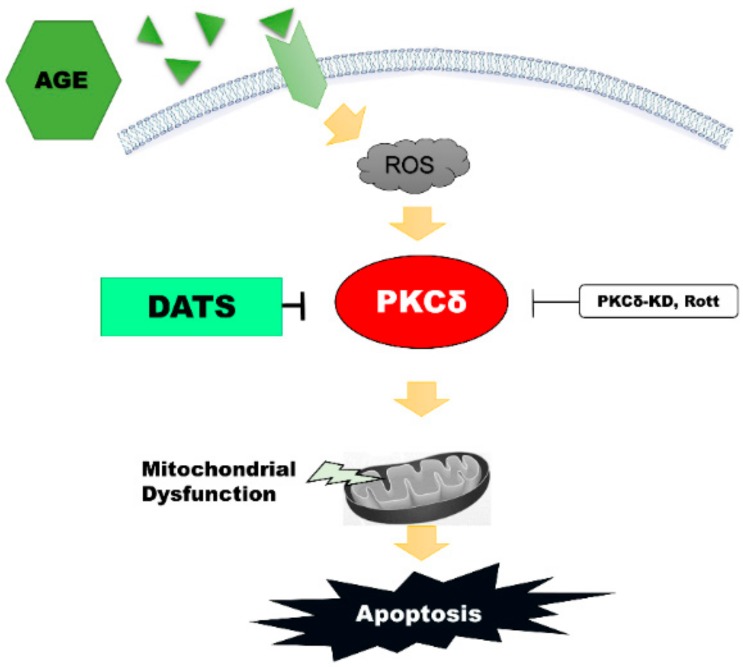
A proposed pathway of DATS attenuating AGE-induced cardiomyocytes apoptosis by inhibiting ROS-mediated PKCδ activation.

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
