# Peer review of "Diallyl Trisulfide (DATS) Suppresses AGE-Induced Cardiomyocyte Apoptosis by Targeting ROS-Mediated PKCδ Activation"

_ijms, 2020, doi:10.3390/ijms21072608_

Round 1
Reviewer 1 Report
The authors didn't respond successfully to all the comments. For example, replicates in the western blots were asked to be included but even though the authors claimed to have included them, there are not shown in the main figures.
Reviewer 2 Report
In the manuscript, the authors investigate how DATS mediate the beneficial effects in the experimental model of diabetic cardiomyopathy, where chronic high glucose levels lead to the production of advanced glycation end products (AGEs).
The manuscript is written well, however, some changes need to be done
- I can recommend the authors to check English in the manuscript (for example, lines 51, 52—“ Furthermore, oxidative stress which were induced by high glucose is defined as the imbalanced of…”, lines 67, 68).
- Fig. 6 the authors show statistical analysis but I do not fully understand the values of the analysis with such a small sample size.
- It is worth discussing whether other pathways could be involved (PI3K, BAD, etc). The works of other researchers on the same topic should be discussed also and the novelty of the present study should be stressed out (Hao, Y.; Liu, H.-M.; Wei, X.; Gong, X.; Lu, Z.-Y.; Huang, Z.-H. Acta Diabetol 2019. Yu, L.; Li, S.; Tang, X.; Li, Z.; Zhang, J.; Xue, X.; Han, J.; Liu, Y.; Zhang, Y.; Zhang, Y.; et al. Apoptosis 2017, 22, 942–954. Yu, L.; Di, W.; Dong, X.; Li, Z.; Xue, X.; Zhang, J.; Wang, Q.; Xiao, X.; Han, J.; Yang, Y.; et al.. Oncotarget 2017, 8, 74791–74805. Yang, H.-B.; Liu, H.-M.; Yan, J.-C.; Lu, Z.-Y. J. Cardiovasc. Pharmacol. 2018, 71, 367–374. Jeremic, J.N.; Jakovljevic, V.L.; Zivkovic, V.I.; Srejovic, I.M.; Bradic, J.V.; Bolevich, S.; Nikolic Turnic, T.R.; Mitrovic, S.L.; Jovicic, N.U.; Tyagi, S.C.; et al.. Mol. Cell. Biochem. 2019, 460, 151–164, etc)
Round 2
Reviewer 1 Report
The authors addressed all the points.
This manuscript is a resubmission of an earlier submission. The following is a list of the peer review reports and author responses from that submission.
Round 1
Reviewer 1 Report
The present manuscript by Ou et al studies the effect of a molecule present in garlic, diallyl trisulfide (DATS), known to have cytoprotective effects. The authors investigate how DATS could mediate potential beneficial effects in the context of diabetic cardiomyopathy. In this pathological condition, chronic high glucose leads to the production of advanced glycation end products (AGEs) resulting in oxidative stress.
The authors claim that DATs treatments reduces ROS and apoptosis through PKC signaling inhibition using cellular models. This results were supported with the use of a diabetic rat model, where DATS treatment reduced PKC signaling and apoptosis markers while increasing a survival marker, AKT.
The manuscript is fairly clear, and the experiments are well described. There are however some issues that should be addressed to strengthen some conclusions.
Overall, the WB from cell culture experiments do not show any replicate, with a single sample it is difficult to interpret the consistency of the results. Fig. 6 shows however duplicates in the mouse experiments and this should be considered a minimum. In fact, the authors show statistical analysis but with a sample size of 2 it is not possible to do statistical analysis.
Due to the lack of replicates, in Fig. 1, the interpretation of AKT is not very conclusive. There doesn’t seem to be a clear decrease of pAKT, particularly in Fig1b.
Fig 1A and 1B at time point 24h and 250 dose, in the pAKT there should be the same difference between control and treatment, this doesn’t seem to be the case. There is no obvious difference in Fig 1A.
Fig 2C, error bar is missing in firs bar.
Total levels of PKC and AKT are missing in several blots, Figure 2E, Fig 4A-B, and Figure 5.
Line 240, figure 9 is not existing.
Line 137, from the western blot of Fig.2E, the effect of AGE on the translocation of PKCδ is not obvious. Total PKCδ is missing.
Line 119, based on the result shown in this section, the author didn’t really prove the inhibitory effect DATS on apoptosis is mediated through blocking PKCδ. Hence, “Inhibitory effect of DATS on AGE-induced cardiac PKCδ-dependent apoptosis” might not be appropriate.
Concerning the signaling experiments, one could wonder whether there are other pathways involved. For example, is PI3K also affected? Or is the phosphorylation of BAD regulated under DATS treatment? This could help better understand and deciphering the molecular mechanisms.
Overall, thanks to the in vivo experiments, the results are more convincing, but it would be even better to explore other phenotypes to further support the effects of DATS in diabetic hearts.
Reviewer 2 Report
Abstract. Please rewrite.
Lines 29-30. Please delete the information not relevant to this study.
Line 29, 31. Please explain the abbreviations in the first time they appear in the text.
Please give the information about experimental models used, number of animals, parameters studied, results obtained.
Introduction.
According to the present form of view “…diabetic cardiomyopathy is defined by the existence of abnormal myocardial structure and performance in the absence of other cardiac risk factors, such as coronary artery disease, hypertension, and significant valvular disease, in individuals with diabetes mellitus” [Borghetti G, von Lewinski D, Eaton DM, Sourij H, Houser SR, Wallner M. Diabetic Cardiomyopathy: Current and Future Therapies. Beyond Glycemic Control. Front Physiol. 2018;9:1514. doi:10.3389/fphys.2018.01514]. With that, the meaning of several sentences is not clear.
Line 52. Please rewrite the sentences of give the reference where the ROS formation is referred as a cause of diabetic cardiomyopathy. To my own opinion, the mechanisms that lead to its development are more complicated.
Line 53-56. Probably, it is not correct to make such assumptions from the results of own experimental work.
Line 56-57. Please rewrite.
Results.
Line 103. Please make the title as a correct sentence.
Line 105-106. Please delete the information not relevant to this part of the manuscript or move it.
Line 119. Please make the title as a correct sentence.
Figure 6. Please add the scale bars and the description.
Discussion
Line 222-223. The sentence is very contradictory.
Line 224-225. Please rewrite.
Line 225-226 “has been proven that 225 aging can cause chronic development of hyperglycemia” please add the reference.
Line 247. “Apoptosis can be divided into intrinsic and extrinsic pathways”. Apoptosis could not be divided. It could be initiated….please rewrite.
Line 248-249. Please give the reference.
Line 284-285. While there is no data about the induction of cardiomyopathy, the conclusion should be rewritten.
Line 363. Animal and treatment protocol.
Please give information about number of animals. Please explain and give the information how the diabetic cardiomyopathy was diagnosed and what was its prevalence. Please give the information, how the end-point of the experiment (16 day) was established.